

# Extraction and characterization of collagen and gelatin from body wall of sea cucumbers *Stichopus horrens* and *Holothuria arenicola*

Noora Barzkar[1,2], Gilan Attaran-Fariman[2], Ali Taheri[3] and Balu Alagar Venmathi Maran[4]

[1] Higher Institution Centre of Excellence, Borneo Marine Research Institute, Universiti Malaysia Sabah, Kota Kinabalu, Sabah, Malaysia
[2] Department of Marine Biology, Faculty of Marine Sciences, Chabahar Maritime University, Chabahar, Iran
[3] Fisheries Department, Faculty of Marine Sciences, Chabahar Maritime University, Chabahar, Iran
[4] Institute of Integrated Science and Technology, Nagasaki University, Nagasaki, Japan

Corresponding author
Noora Barzkar,
noora.barzkar@ums.edu.my

## ABSTRACT

**Background:** Marine invertebrates, including sponges, molluscs, jellyfish, mussels, and sea cucumbers, are abundant sources of high-quality collagen and offer advantages such as availability, ease of processing, lower inflammatory response, and good metabolic compatibility. Approximately 70% of the total protein in the body wall of sea cucumbers is collagen. Gelatin is a water-soluble protein produced from heat-denatured collagen and has various industrial applications.

**Methods:** Pepsin-solubilized collagen was extracted from the body wall of two sea cucumber *Stichopus horrens* and *Holothuria arenicola*, species found in the Oman Sea and characterized with SDS-PAGE and amino acid composition. Then gelatin was extracted from pepsin-solubilized collagen of *S. horrens* and some rheological properties were measured.

**Results:** Amino acid composition and SDS-PAGE analysis showed that the collagen from both species was type I, with one α1 chain and β chains, with molecular weights of 125 and 250 kDa, respectively. Glycine was the most abundant amino acid in the collagen from both sea cucumber species. The pepsin-soluble collagens from both species had high levels of glycine, proline, alanine, glutamic acid, and hydroxyproline. The gelatin from *S. horrens* had a melting point of 30 °C and displayed exceptional thermal stability, surpassing that of mammalian gelatin. Its gelling point was 5 °C, like that of cold-water fish gelatin, with a viscosity of 2.065 cp-lower than mammal gelatins. These findings suggested that collagen and gelatin from sea cucumbers could be useful in nutraceutical, pharmaceutical and cosmetic industries.

## INTRODUCTION

Natural and bioactive compounds from marine organisms are rich sources of nutritional, pharmaceutical, and medicinal compounds (*Barzkar et al., 2024*; *Shah et al., 2022*; *Taheri & Bakhshizadeh, 2020*). Sea cucumbers belong to the phylum Echinodermata, class Holothuroidea, and are commonly called holothurians. Currently, there are approximately 1,400 species of holothurians worldwide (*WoRMS, 2024*). The FAO has reported 58 species of commercial importance that are harvested and commonly exploited in more than 70 countries, although many other species are exploited in relatively small quantities (*Purcell et al., 2023*). Sea cucumbers are used for dietary and medicinal purposes in Asian countries because of their chemical composition (wet or dried forms) (*Bordbar, Anwar & Saari, 2011*). The main edible part of sea cucumbers is the body wall, which mostly consists of collagen and mucopolysaccharides; and approximately 70% of the total body wall protein consists of highly insoluble type I collagen fibers (*Barzkar et al., 2023a*).

Collagens are the main structural proteins of multicellular animals in connective tissues, including skin, bone, teeth, blood vessels, intestines, and cartilage, making up about 30% of the total protein (*Di Lullo et al., 2002*; *Müller, 2003*). Collagen is a long rod-like molecule that consists of three polypeptide α-chains with predominantly repeating Gly-X-Y triplets (X and Y are frequently proline and hydroxyproline), which cause each α-chain to form a left-handed helix with distinct primary, secondary, tertiary, and quaternary structures (*Eckhard et al., 2014*; *Fields, 2013*; *Kadler et al., 2007*; *Pal, Nidheesh & Suresh, 2015*). Collagen contains 20 amino acid residues that play a key role in identifying various types of collagen. So far, 29 different types of collagen have been identified in vertebrate tissues, with each type differing significantly in structure, amino acid sequence, and function, likely related to a specific genetic variant (*Barzkar, Sukhikh & Babich, 2024*; *Henriksen & Karsdal, 2024*).

The classification of collagens is based on the expression of various genes during tissue construction. Collagen type I is the most abundant and economically important type found in various tissues, such as the heart, tendons, skin, bones, lungs, corneas, and blood vessels (*Amirrah et al., 2022*; *Barzkar et al., 2023b*). Type I collagen is extracted from sponges, jellyfish, crustaceans, and sea cucumbers (*Abedin et al., 2013*; *Cui et al., 2007*; *Saito et al., 2002*; *Venmathi Maran et al., 2023*). Owing to its importance and unique characteristics, collagen is widely used in cosmetic, biomedical, pharmaceutical, and food industries (*Sohail et al., 2022*; *Barzkar et al., 2022*; *Barzkar, Jahromi & Vianello, 2022*; *Kittiphattanabawon et al., 2005*; *Neklyudov, 2003*). Gelatin is a water-soluble protein produced from heat-denatured collagen that has traditional applications in the food, photography, cosmetics, and pharmaceutical industries. Recently, there have been a growing number of novel uses of gelatin in the food industry as emulsifiers, colloid stabilizers, fining agents, biodegradable packaging materials, and microencapsulating agents, aligning with the trend of replacing synthetic agents with more natural ones (*Said & Sarbon, 2022*). The major sources of collagen and gelatin are porcine and bovine skin and bone (*Kord et al., 2024*; *Sukhikh et al., 2020*).

However, there is a need to develop alternative collagen sources, leading to the extraction of collagen from marine organisms, including sponges, jellyfish, mussels, and sea cucumbers, which are abundant sources of high-quality collagen and, offer advantages such as availability, ease of processing, lower inflammatory response, and good metabolic compatibility.

The body wall of sea cucumbers has been proposed as a potential source of collagen that does not pose the risk of endemic diseases, such as transmissible spongiform encephalopathy (TSE), bovine spongiform encephalopathy (BSE), and foot and mouth disease (FMD), offering high yields and availability (*Barzkar et al., 2021*; *Panggabean et al., 2023*; *Senadheera, Dave & Shahidi, 2020*). Moreover, according to *Senadheera, Dave & Shahidi (2020)* collagen derived from fishery products has greater thermostability and is tightly packed. Collagen is the main bioactive compound extracted from sea cucumbers and is a sustainable and green source of native fibrillary collagen for the production of thin membranes for regenerative biomedical applications (*Senadheera, Dave & Shahidi, 2020*).

Information on gelatin from sea cucumber species is limited, with reports on collagen from some sea cucumber species, including *Stichopus japonicus* (*Saito et al., 2002*), *Cucumaria frondosa* (*Trotter et al., 1995*), *Stichopus vastus* (*Abedin et al., 2013*), and *Holothuria parva* (*Adibzadeh et al., 2012*). However, there are no reports on the properties of collagen and gelatin from *Stichopus horrens* Selenka, 1867 and *Holothuria arenicola* Semper, 1868 which are abundant in some localities in the Western Pacific, parts of Asia and the Indian Ocean.

This study aimed to extract crude collagen fibrils and isolate pepsin-soluble collagen, and collagen-derived gelatin from the body wall of two sea cucumber species, *S. horrens* and *H. arenicola*. Furthermore, we determined the amino acid composition of collagen and the rheological properties of gelatin from sea cucumber collagen and collagen-derived gelatin, with the goal of proposing them as alternative marine sources for industrial uses and adding value to sea cucumber species.

## MATERIALS AND METHODS

### Sampling and preparation of sea cucumber's body wall

Sea cucumber samples (*S. horrens* and *H. arenicola*) were gathered from ranging about 10 m depths by scuba diving from Chahbahar Bay located in the Southeast coast of Iran and North part of the Oman Sea. The samples were then placed on ice and transported to the laboratory. Wet weight 523.45 g of *S. horrens* and 738.63 g of *H. arenicola* were used for the experiment. They were cleaned by washing with cold water and kept at −20 °C for a week. The body walls of both species were cut into small pieces (approximately 1 cm × 1 cm).

### Extraction of crude collagen fibrils

Collagen from the body walls of *S. horrens* and *H. arenicola* was extracted following the method (*Kim et al., 2012*). The body wall fragments (100 g wet weight) were homogenized in 1,000 cc of distilled water for 0.5 h, the water was changed, and the extraction process was repeated twice for 30 min. Then, the water was replaced with 1,000 cc of 0.1 M Tris–HCl, 4 mM EDTA, pH 8.0, and gently stirred for 72 h. The mixture was then replaced

with 600 cc of distilled water and gently homogenized for 48 h. The solution was centrifuged at 11,000×$g$ for 15 min and the supernatant was again centrifuged at 8,000×$g$ for 1 h. The supernatant was freeze-dried to obtain collagen. The lyophilized product was crude collagen fibers. The dried crude collagen fibers were collected and stored in a refrigerator.

## Isolation of pepsin-solubilized collagen (PSC)

The following steps were performed using the method described by *Saito et al. (2002)*. The crude collagen fibrils obtained from both samples were stirred in 20 (v/w) of 0.1 M sodium hydroxide for 72 h, then cleaned with distilled water and homogenized with 10 (v/w) of 0.5 M acetic acid consisting of porcine pepsin at a ratio of 1:100 (w/w) for 3 days. The PSC in the supernatant salted out by adding 0.8 M NaCl. The precipitate was centrifuged at 8,000×$g$ for 30 min, collected, suspended in 0.5 M acetic acid and dialyzed against 0.02 mol/L $Na_2HPO_4$ (pH 8.0) for 48 h to inactivate pepsin and lyophilized. Dried and purified PSC were stored in a refrigerator (*Saito et al., 2002*).

## Sodium dodecyl sulfate poly-acrylamide gel electrophoresis (SDS-PAGE)

Protein patterns of collagen samples from the two species of sea cucumber in this study were analyzed using SDS-PAGE, as previously described (*Laemmli, 1970*). The samples were mixed in 0.1M $Na_3PO_4$ (pH 7.2) and dissolved in a sample loading buffer (60 mmol/l Tris–HCl, pH 8.0, containing 0.1% bromophenol blue, 2% SDS, 25% glycerol) at a 4:1 (v/v) ratio. Electrophoresis was performed on 9% polyacrylamide gels. Each gel was stained with 0.05% (w/v) Coomassie Brilliant Blue (R250) dissolved in 45% (v/v) methanol and 10% (v/v) acetic acid, and destained with a solution containing methanol and acetic acid (3:1, v/v).

## Determination of amino acid composition

Lyophilized samples of PSC from the two species (10 mg) were hydrolyzed in a gas tube using 6 N HCl at 110 °C for 24 h. Amino acids were analyzed by high-performance liquid chromatography (HPLC), using a C18 column, prior to derivatization with ninhydrin, and measured at 570 nm, except for proline and hydroxyproline, which were measured at 440 nm (*Gilbert & Townshend, 1987*). The amino acid content is presented as the number of residues per 1,000 residues.

## Extraction of collagen-derived gelatin from the body wall of *S. horrens*

Gelatin was extracted following the method presented by *Gudmundsson & Hafsteinsson (1997)* from body wall and washed with distilled water to eliminate extra material. The samples were then sequentially treated with 0.2% (w/v) NaOH, 0.2% $H_2SO_4$ and 1.0% citric acid for 30 min. After each treatment, the body walls were washed with water until a neutral pH was reached, before adding a new solution. Finally, the body walls were washed with distilled water. Gelatin extraction was performed using three volumes of distilled water at 45 °C for 12 h. The supernatant was filtered through Whatman filter paper (no. 1). The filtrate was lyophilized and stored in a refrigerator (*Gudmundsson & Hafsteinsson, 1997*).

## Determination of gelatin yield

The optimized gelatin yield was calculated based on *Balaji Wamanrao & Tanaji (2022)* by some modification using the following formula:

*Yield of gelatin* $(\%) = (C \times V)/M \times 100$

    C (mg/ml) = Concentration of gelatin in sample solution
    V (Laemmli) = Total volume of extracted solution containing gelatin
    M (mg) = Primary weight of body wall sample

## Protein concentration assay

The total protein content of gelatin samples from *S. horrens* was determined using the biuret method (*Gornall, Bardawill & David, 1949*).

## Rheological properties of gelatin from sea cucumber collagen and collagen-derived gelatin

### Gelling point and gelling time determination

The setting point and time were determined using the following method (*Muyonga, Cole & Duodu, 2004*). A 10% (w/v) gelatin solution was prepared, and 30 ml was transferred to an exam pipe and placed in a hot water bath at 40 °C. The water bath was then cooled instantly in ice-chilled water at 2 °C. The thermometer was placed in the solution and removed every 15 s until any drop did not drip; this temperature and gelatin setting point were recorded and noted.

### Melting point and melting time determination

The melting point and melting time of the gelatin were determined (*Muyonga, Cole & Duodu, 2004*). Gelatin solution at 10% (w/v) was prepared and placed in a freezer at −7 °C for 16–18 h. The frozen sample was then transferred to a water bath at 10 °C, warm water was slowly added until the final temperature reached 45 °C, and the melting temperature and time were recorded.

### Viscosity determination

Viscosity measurements of the gelatin samples were carried out according to the British Standard Institution method (*British Standards Institutions, 1975*). Gelatin samples were obtained by dissolving lyophilized gelatin in distilled water at 40 °C for 15–20 min. The dispersions were then transferred to a capillary viscometer (No. 150; Cannon–Fenske) to determine the viscosity. The viscometer was placed in a hot water bath at 60 °C for 10 min with steady temperature and the return time of lines B to A was recorded using a chronometer. The viscosity was calculated according to $V = k (t-\theta)$ equation. The viscosity of the gelatin samples is expressed as centipoise (cP).

## RESULTS

### Electrophoresis (SDS-PAGE)

The PSC from the body walls of two sea cucumber species, *S. horrens* and *H. arenicola* was analysed using SDS-PAGE with a 9% polyacrylamide gel (Fig. 1). The electrophoretic

profile revealed two collagen chains: one major profile ($\alpha_1$) with an apparent molecular weight of around 125 kDa, and the other ($\beta$) with a small number of dimers and an apparent molecular weight of approximately 250 kDa. This indicated that the primary component of collagen extracted from *S. horrens* and *H. arenicola* could be classified as collagen type I, consisting of $\alpha$ and $\beta$ chains.

## Amino acid composition of PSCs derived from *S. horrens* and *H. arenicola*

The amino acid composition of the body wall pepsin-soluble collagen of the two species of sea cucumbers is shown in Fig. 2. Glycine was found to be the most common amino acid in collagen of both species. PSCs from both species contained high levels of glycine, proline, alanine, glutamic acid, and hydroxyproline. Both species of sea cucumbers had high imino acid content (proline + hydroxyproline) in their PSC. No cysteine was found in the amino acid collagen compounds in the body walls of either sea cucumber species.

## Ratio of hydroxyproline to proline

Table 1 presents the proline/hydroxyproline and collagen ratios in the sea cucumbers *S. horrens* and *H. arenicola*. The data indicate that the ratio of proline/hydroxyproline were 0.70 and 0.75 in *S. horrens* and *H. arenicola*, respectively.

## Ratio of lysine to arginine

The lysine/arginine ratios in the collagen of *S. horrens* and *H. arenicola* are shown in Table 1. The ratio of proline/hydroxyproline was 0.14 and 0.13 in *S. horrens* and *H. arenicola*, respectively.

## Yield of gelatin extraction from *S. horrens* and chemico-physical properties

The yield of gelatin extracted from the body walls of *S. horrens* was approximately 1.5% for 100 g of skin. The chemico-physical properties of gelatin extracted from the body wall of *S. horrens* are provided in Table 2, indicating a high protein concentration, high melting point, and low gel formation temperature. It is important to mention that, because of the lack of access to *H. arenicola*, no study on its gelatin has been conducted.

# DISCUSSION

## Extraction of collagen

Collagen tissue is typically extracted and isolated using either organic acids (such as acetic acid) or inorganic acids (like hydrochloric acid). In 1973, Matsumura employed beta-mercaptoethanol (B-ME) to solubilize collagen with pepsin from the body wall of the sea cucumber *S. japonicus* (*Matsumura, 1973*). Subsequently, *Trotter et al. (1995)* developed a method that omitted B-ME, enhancing the production efficiency of collagen fibers. They also used ionized EDTA solution to dissociate skin tissue (*Trotter et al., 1995*). Typically, pepsin digestion of collagen occurs in 0.5 M acetic acid at a low temperature, yielding pepsin-soluble collagen. The addition of pepsin in acid can boost collagen

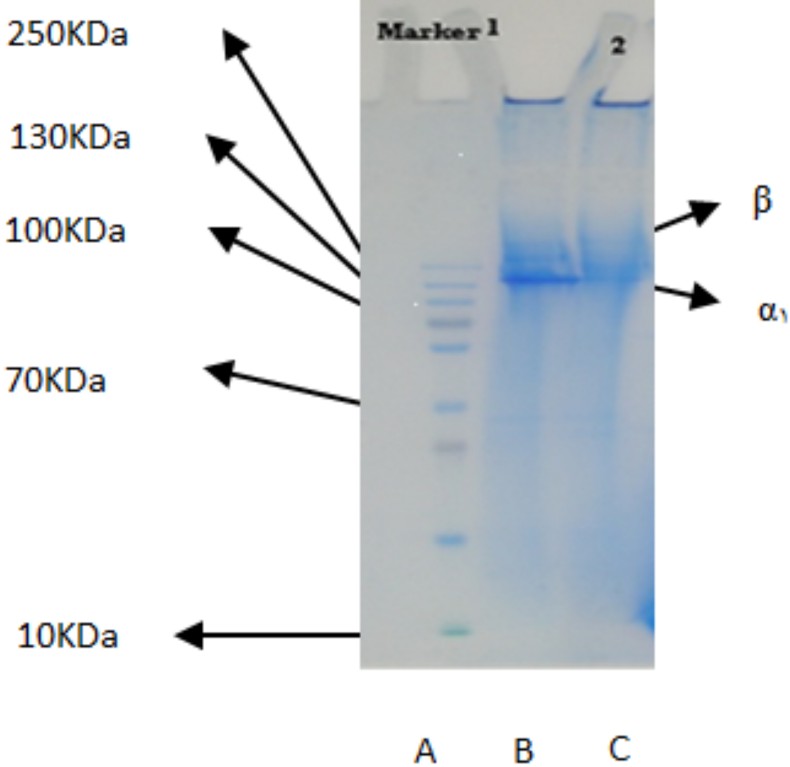

**Figure 1 SDS-polyacrylamide-gel electrophoresis of collagen from the body wall of *S. horrens* and *H. arenicola*.** Lane A: high molecular weight marker; Lane B: PSC (pepsin solubilized collagen from *S.horrens*); Lane C: PSC (pepsin solubilized collagen from *H. arenicola*).

productivity by increasing its solubility in 0.5 M acetic acid (*Trotter et al., 1995*). Covalent cross-linking of collagen molecules, influenced by the density of aldehyde groups in the telopeptide region, can reduce collagen solubility (*Wu et al., 2014*).

*Park et al. (2012)*, suggested that pepsin-soluble collagen could be a viable alternative to mammalian collagen in the pharmaceutical and nutraceutical industries. Based on these findings, this study involved homogenizing crude extracted collagen fibrils with 0.5 M acetic acid and porcine pepsin for 3 days to enhance the solubility of the acid-extracted collagen (*Park et al., 2012*).

## Electrophoresis (SDS-PAGE)

The electrophoresis patterns of $\alpha_1$ and $\beta$ chains of soluble collagen in pepsin from two sea cucumber species studied were similar to those reported for other sea cucumber species (*Li et al., 2020*). These findings indicate that the primary component of collagen extracted from *S. horrens* is type I collagen, consisting of $\alpha_1$ and $\beta$ chains. This is consistent with collagen patterns observed in collagens from other sea cucumber species including *S. japonicas* (*Cui et al., 2007*), *Parastichopus californicus* (*Liu, Oliveira & Su, 2010*), *Stichopus vastus* (*Abedin et al., 2013*; *Bechtel et al., 2013*), *Bohadschia* spp. (*Yd et al., 2013*), and *Holothuria cinerascens* (*Li et al., 2020*).
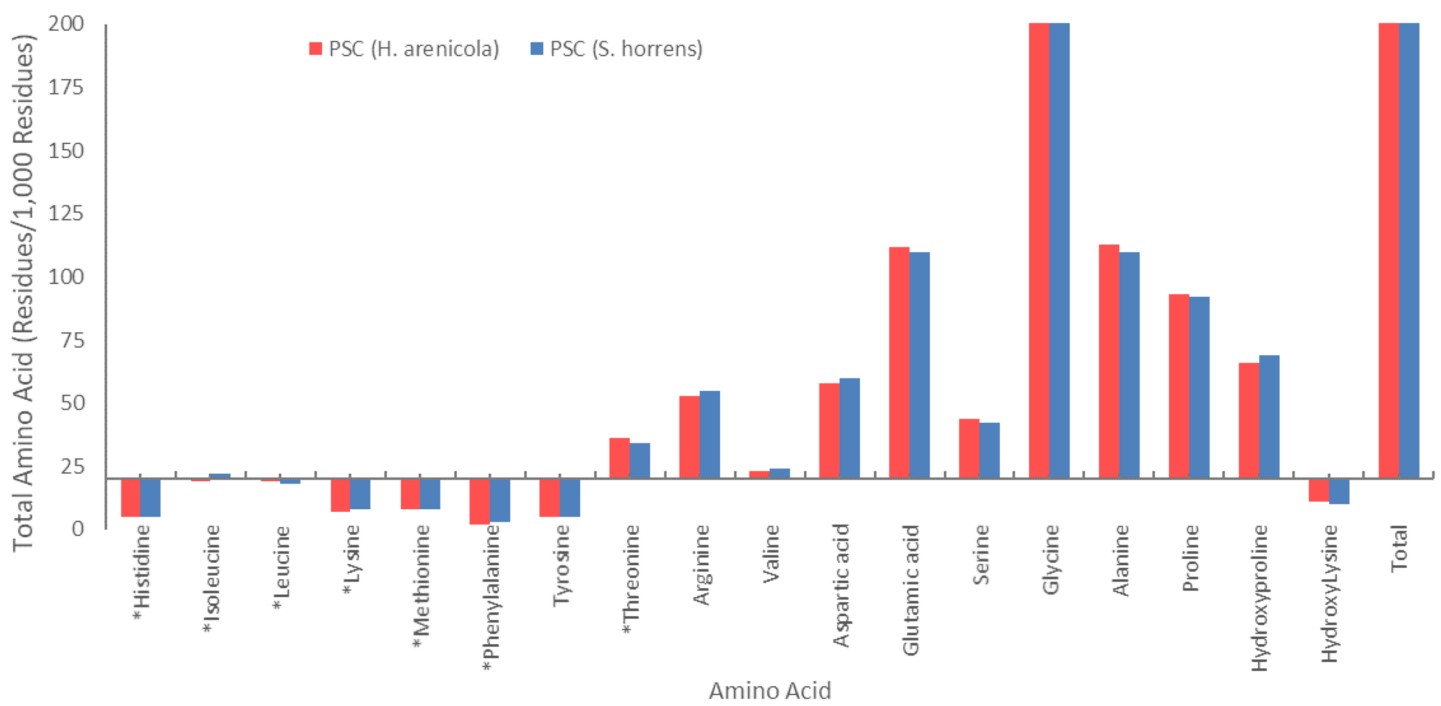

**Figure 2 Amino acid composition of collagens isolated from *S. horrens* and *H. arenicola* (Residues/1,000 residues).**

**Table 1 Ratio of proline/hydroxyproline and lysin/argenine in collagen of sea cucumber *S. horrens* and *H. arenicola*.**

| Ratio | *H. arenicola* | *S. horrens* |
|---|---|---|
| Proline/hydroxyproline | 0.75 | 0.70 |
| Lysin/argenine | 0.13 | 0.14 |

**Table 2 Chemicophysical properties of the gelatin from body wall of sea cucumber *S. horrens*.**

| Property | Value |
|---|---|
| Protein concentration (mg/ml) | 87.93 |
| Gel melting time (min) | 10 |
| Gel melting point (°C) | 30 |
| Gelling time (min) | 8 |
| Gelling Point (°C) | 5 |
| Viscosity (60 °C) (cP) | 2.065 |

The collagen found in the body wall of the sea cucumbers in this study has a high molecular weight of 125–250 kDa, which is higher than sponge collagen (58 kDa) and bovine bone collagen (25.3–11.7 kDa), but lower than porcine skin collagen (205–150 kDa) (*Li, Jia & Yao, 2009*). SDS-PAGE was used to detect collagen subunits and identify the

presence of α chain, particularly $\alpha_1$ and $\alpha_3$. This method is fast and effective for determining collagen structure and type (*Davis et al., 2001*; *Tye, Hunter & Goldberg, 2005*). Variations in α chains lead to different types of collagen. Collagen type I typically consists of a heterotrimer with two different subunits, $\alpha_1$ and $\alpha_2$. The usual chain composition of collagen is $(\alpha_1)_2\alpha_2$, but it can be also $\alpha_1\alpha_2\alpha_3$. The presence or absence of the $\alpha_3$ chain in the body wall collagen of *S. horrens* and *H. arenicola* remains uncertain because the $\alpha_3$ chain separated by SDS-PAGE has a similar chemical structure and migration characteristics to the $\alpha_1$ chain.

The collagen chains of *S. horrens* and *H. arenicola* showed results similar to the collagen extracted from the skin of *Sepia lycidas*, with a single alpha bond and a prominent β band in soluble pepsin collagen. Previous reports on PSC extracted from *S. japonicus* and *P. californicus* indicated these collagens contain a homotrimer with molecular weights of 135 and 138 kDa, respectively (*Cui et al., 2007*; *Zhang, Liu & Li, 2009*), which is higher than the molecular weight observed in this study. Additionally, *Stichopus vastus* has 122 KDa ($\alpha_1$) and 267 KDa (β dimers), *Australostichopus mollis* Contain 267 KDa (β dimers) and small amount of γ components beside ($\alpha_1$ $\alpha_2$ chains) and *Holothuria cinerascens* 80–90 KDa (α chains) and 150–160 KDa (β-chain) (*Abedin et al., 2013*; *Li et al., 2020*; *Liu et al., 2017*). Collagen from *S. japonicus* sea cucumber species has subunits $(\alpha_1)_3$ (*Cui et al., 2007*), while *C. frondosa* contains subunits $\alpha_1$ and β (*Trotter et al., 1995*). In sea urchin, reported collagen contains subunits $(\alpha_1)_2\alpha_2$ (*Omura, Urano & Kimura, 1996*), whereas the collagens from both sea cucumber species in this study have subunits $\alpha_1$ and β. Collagens from many species, including *Yezo Sika* tendon, have a heterotrimer compound of peptic-soluble collagen with subunit $(\alpha_1)_2\alpha_2$.

Collagen combinations, including the heterotrimer $(\alpha_1)_2\alpha_2$ are commonly found in various marine species such as carp scales (*Zhang et al., 2011*), silver carp (*Rodziewicz-Motowidło et al., 2008*), *S. japonicus* (*Saito et al., 2002*), catfish (*Bama et al., 2010*), octopus (*Kimura, Takema & Kubota, 1981*), pearl oyster (*Mizuta et al., 2002*). The skin of octopus (*Kimura, Takema & Kubota, 1981*), squid (*Mizuta et al., 1994*; *Shadwick, 1985*), red sea bream (*Nagai, Izumi & Ishii, 2004*), and Japanese common star fish (*Lee et al., 2009*) also contain collagen chain combinations of the heterotrimer $(\alpha_1)_2\alpha_2$, indicating a similarity to mammalian collagen, such as porcine skin. Collagen type I found in the skin of puffer fish (*Nagai, Araki & Suzuki, 2002*), octopus skin (*Kimura, Takema & Kubota, 1981*), jellyfish (*Venmathi Maran et al., 2023*; *Yuliya Kulikova et al., 2024*), Japanese red sea bream scales (*Nagai, Izumi & Ishii, 2004*), purple sea urchin (*Nagai & Suzuki, 2000*), Japanese common star fish (*Lee et al., 2009*) is similar to mammalian collagen, including porcine collagen. This study suggests that the collagen composition of both sea cucumber species is comparable to that of various marine species and mammals reported globally.

## Amino acid composition

Amino acid compositions are crucial in distinguishing different types of collagens, with glycine being a notable feature. The extracted collagen from *S. horrens* and *H. arenicola* in this study contains eighteen amino acids, including nine essential amino acids (EAAs) and

nine non-essential amino acids (NEAAs). The ratio of EAAs to NEAAs reflects the protein quality of the collagen, which, in this work, was found to be equal.

Glycine is a key component of collagen molecules. In Fig. 2, the amino acid compositions of the PSCs from *S. horrens* and *H. arenicola* showed a high glycine content as major non-essential amino acids, which accounted for the third of the total amino acid residues (325 residues/1,000) in *S. horrens* species and (326 residues/1,000) in *H. arenicola* species. Similar result reported for *S. japonicus* (Cui et al., 2007), *H. tubulosa* and *H. polii* (Sicuro et al., 2012). *Holothuria scabra* and *Actinopyga mauritiana* (Omran, 2013), *H. arenicola* and *A. mauritiana* (Haider et al., 2015), *S. vastus* (Rasyid, 2018), *H. scabra* (Saallah et al., 2021), and *Stichopus chloronotus* (Xia et al., 2022). It is reported that the formation of the tri-peptide unit in collagen is due to the presence of glycine at every third residue of the helix. Stabilization of the collagen helix is based on the content of proline and hydroxyproline in the tri-peptide sequence (Jamilah et al., 2013). Maintaining the health of central nervous and digestive human system and antioxidant properties for cancer protection by inhibition of inflammation and chemoprevention of carcinoma are reported as functional properties of the glycine (Ridzwan et al., 2014; Wen, Hu & Fan, 2010). Therefore, high levels of the glycine in the collagens of this study will be useful in pharmaceutical industries.

Alanine, glutamic acid, proline, and hydroxyproline were also present in significant amounts. The amino acid content of alanine and glutamic acid was 110 residues/1,000, proline was 92 residues/1,000, and hydroxyproline was 69 residues/1,000 in the *S. horrens* species, while the tyrosine, phenylalanine, and histidine contents of PSC in the two species were low. The amino acid content of alanine, glutamic acid, proline, and hydroxyproline in the *H. arenicola* species was 113, 112, 93, and 66 residues per 1,000 residues, respectively. PSC from the two studied sea cucumber species of the current study has high levels of alanine, hydroxyproline, proline, and glutamic acid. The significant presence of these amino acids suggest that collagen is the major protein structure in the body wall of these sea cucumbers (Cui et al., 2007; Saito et al., 2002).

The amount of proline plus hydroxyproline in the hydrolyzed collagen was 151 units per thousand in *H. arenicola* species and 161 units per thousand in *S. horrens*. The latter value is approximately equal to the pepsin-soluble amino acid content reported by Saito et al. (2002) and Cui et al. (2007), and is higher than the proline plus hydroxyproline content of PSC from *H. arenicola* (159 units per thousand) observed in this study (Cui et al., 2007; Saito et al., 2002). However, it is lower than the proline plus hydroxyproline content (186 units per thousand) content of PSC from the skin of unicorn leatherjacket (*Aluterus monoceros*) and carp. The high levels of hydroxyproline in the collagen of both sea cucumber species studied are significant, as they influence the functional properties of collagen. Hydroxyproline contributes to the formation of hydrogen bonds through hydroxyl groups and enhances the thermal stability of collagen (Silva et al., 2014).

The hydroxyproline/proline ratio is important for identifying the type of collagen present. In this study the ratio was 0.75 and 0.70 for *H. arenicola* and *S. horrens*, respectively. These ratios are similar to those reported for other sea cucumber species, such as *H. scabra* (0.77), *S. japonicus* (0.77), *P. californicus* (0.7), and *H. parva* (0.61) have been

reported (*Adibzadeh et al., 2012*; *Cui et al., 2007*; *Saallah et al., 2021*). These results illustrate that the structure of PSC from *H. arenicola* is closer to type I collagen than *S. horrens* (*Liu, Oliveira & Su, 2010*).

Arginine content was 55 residues/1,000 in *S. horrens* and 53 residues/1,000 in *H. arenicola*, similar to findings in *S. chloronotus* (*Xia et al., 2022*). Arginine is essential in collagen synthesis, with functions in cell regeneration, delaying aging, improving brain function, and enhancing immunity (*Xia et al., 2022*). Moreover, literature indicates that a low lysine-to-arginine ratio is associated with reduced plasma cholesterol levels (*Bordbar, Anwar & Saari, 2011*; *Kravitz, Gaisler-Salomon & Biegon, 2013*). In this study, the ratio was 0.13 in *H. arenicola* and 0.14 in *S. horrens*. These values are similar to those found in *H. scabra* (0.13), *H. fuscogilva* (0.13) and *H. fuscopunctata* (0.15), but lower than in *Thelenota ananas* (0.39), *Thelenota anax* (0.33), *A. mauritiana* (0.25), *B. argus* (0.36) and *Apostichopus japonicus* (0.62) (*Wen, Hu & Fan, 2010*; *Yang, Hamel & Mercier, 2015*). Given that a low lysine-to-arginine ratio contributes to hypocholesterolemic effects, the findings suggest that collagen from *S. horrens* and *H. arenicola* may also have such properties.

The amino acid compositions of the body wall collagen from two sea cucumber species were compared with calfskin, porcine, and *S. japonicus* collagens (Table 3). The results indicate that histidine levels in *S. horrens* and *H. arenicola* collagen are similar to those in bovine skin, but lower than in porcine skin. The levels of glycine, alanine, proline, hydroxyproline, lysine, and leucine in collagen extracted from *S. horrens* and *H. arenicola* are lower compared to those in calf and porcine collagen. In contrast, hydroxylysine, serine, glutamic acid, aspartic acid, valine, arginine, threonine, tyrosine, methionine, and isoleucine levels in these sea cucumber species are higher than in porcine and bovine collagen. The phenylalanine content is similar in calf and *S. horrens* sea cucumber collagen, but higher in porcine collagen compared to the two sea cucumber species. Overall, the amino acid composition and distribution of *S. horrens* and *H. arenicola* collagen are similar to calfskin collagen. Additionally, histidine levels in PSC from both sea cucumber species are lower than those in PSC from *S. japonicus*, as are the levels of proline and hydroxyproline.

The amounts of amino acids of glycine, alanine, glutamic acid, valine, arginine, tyrosine, threonine, lysine, leucine, and isoleucine in collagens from *S. horrens* and *H. arenicola* is slightly higher than those in collagen extracted from *S. japonicus*. However, the level of aspartic acid and phenylalanine are higher in the collagen from *S. japonicus* compared to both sea cucumber species studied.

The nutritional value of the protein is assessed by comparing its amino acid content with the EAA profile of a reference protein, typically used for children aged 2–5 years as the standard for all age groups except geniuses and talents. Table 4 presents the EAA profile of collagen from *S. horrens* and *H. arenicola*. It shows that the threonine content in collagen from *S. horrens* meets the requirement for children, while that in *H. arenicola* exceeds it. Overall, the EAA levels in collagen from both sea cucumber species are lower than those needed by children, though valine, isoleucine, and methionine levels are

**Table 3 Comparison of the amino acid composition of the collagen from body wall of sea cucumber _S. horrens_ and _H. arenicola_ of the current study with calf skin, porcine and collagen from body wall of sea cucumber _Stichopus japonicus_.**

| Amino acid | _H. arenicola_ | _S. horrens_ | Porcine (_Ikoma et al., 2003_) | Calf (_Li et al., 2004_) | _S. japonicas_ (_Cui et al., 2007_) |
|---|---|---|---|---|---|
| His | 5 | 5 | 7 | 5 | 2.2 |
| Ile | 19 | 22 | 10 | 11 | 15 |
| Leu | 19 | 18 | 22 | 23 | 15.9 |
| Lys | 7 | 8 | 27 | 26 | 6.4 |
| Met | 8 | 8 | 6 | 6 | – |
| Phe | 2 | 3 | 12 | 3 | 6 |
| Tyr | 5 | 5 | 1 | 3 | 4.8 |
| Thr | 36 | 34 | 16 | 18 | 33.9 |
| Arg | 53 | 55 | 48 | 50 | 50.8 |
| Val | 23 | 24 | 22 | 21 | 19.2 |
| Asp | 58 | 60 | 44 | 45 | 62.7 |
| Glu | 112 | 110 | 72 | 75 | 105.2 |
| Ser | 44 | 42 | 33 | 33 | 42.7 |
| Gly | 326 | 325 | 341 | 330 | 324.7 |
| Ala | 113 | 110 | 115 | 119 | 97.5 |
| Pro | 93 | 92 | 123 | 121 | 107.1 |
| Hyp | 66 | 69 | 97 | 94 | 84.4 |
| Hyl | 11 | 10 | 7 | 7 | – |
| Total | 1,000 | 1,000 | 1,000 | 1,000 | 1,000 |

comparable to those required for adults. Other EAAs are also lower than expected for adults.

Both sea cucumber species lack essential amino acids such as tryptophan and cysteine, and their collagen is deficient in several other EAAs. Consequently, collagen alone does not offer high nutritional value. Diets deficient in essential amino acids may negatively impact health and growth in young children. Collagen's nutritional quality can be improved by combining it with proteins rich in missing amino acids. Despite its limitations as a primary dietary protein, PSC has demonstrated various bioactive properties, including antioxidant (_Lee et al., 2021_), ACE inhibitory (_Zhou et al., 2021_), Immunomodulatory (_Yun et al., 2022_), Antifatigue (_Yu et al., 2021_), Neuroprotective (_Zhao et al., 2022_), Antiosteoporosis (_Yun et al., 2022_), Anti-inflammatory (_Zhang et al., 2021_) and Antiaging (_Guo et al., 2020_) effects. Therefore, while collagen from sea cucumbers may not be ideal as a standalone nutritional component, it holds potential in nutraceutical and pharmaceutical applications (_Man et al., 2023_). Further preclinical and clinical testing is necessary to fully explore its benefits.

## The yield of gelatin extraction

The gelatin yield in this study was lower compared to the range reported for fish gelatin, which is (6–19% gelatin dry weight/wet skin) (_Karim & Bhat, 2009_). Although there is no

**Table 4 Patterns of essential amino acids collagen from body wall of sea cucumber *S. horrens* and *H. arenicola* and essential amino acids in a daily diet.**

| AA | EAA in children | EAA in minors | EAA in adults | Children (Cs H)* | Minors (Cs H)* | Children (CsS)* | Minors (CsS)* | Adults (CsS)* | Adults (CsH)* |
|---|---|---|---|---|---|---|---|---|---|
| His | 19 | 26 | 10 | 26.31 | 19.23 | 26.3 | 19.2 | 50 | 50 |
| Ile | 28 | 46 | 20 | 67.85 | 41.3 | 78.57 | 47.8 | 110 | 95 |
| Leu | 66 | 93 | 39 | 28.78 | 20.43 | 27.27 | 19.35 | 62.06 | 48.71 |
| Lys | 58 | 66 | 30 | 12.06 | 10.6 | 13.79 | 12.12 | 26.66 | 23.33 |
| Met + Cys | 25 | 42 | 10 | 32 | 19.04 | 32 | 19.04 | 80 | 80 |
| Phe + Tyr | 63 | 72 | 25 | 11.11 | 9.72 | 12.69 | 11.11 | 32 | 28 |
| Trp | 11 | 17 | 4 | 0 | 0 | 0 | 0 | 125 | 125 |
| Thr | 34 | 43 | 15 | 105.88 | 83.72 | 100 | 79.06 | 226.6 | 240 |
| Val | 35 | 55 | 26 | 65.71 | 41.81 | 68.57 | 4.63 | 92.3 | 88.46 |
| Total | 320 | 434 | 179 | 349.7 | 245.85 | 391.19 | 251.31 | 824.4 | 778.5 |

**Note:**
(AA)*, Amino acid; (EAA)*, Essential amino acid; (CsH)*, Chemical index of H. arenicola; (CsS)*, Chemical index of S.horrens.

data available on gelatin yield from other sea cucumber species, the yields for various fish species such as *Clarias gariepinus* (5.85%), *Oreochromis nilotica* (7.81%), *Oreochromis mossambicus* (5.39%), *Saurida tumbil* (10.74%) and *Thunnus albacares* (18%) are higher than that observed in this study (*Biluca, Marquetti & da Trindade Alfaro, 2011*; *Jamilah & Harvinder, 2002*; *Rahman, Al-Saidi & Guizani, 2008*; *Taheri et al., 2009*). According to our previous report, the body wall of *S. horrens* has a high moisture content of 92.8% (*Barzkar, Attaran Fariman & Taheri, 2017*), which is higher than that of fish skin and may contribute to the lower gelatin yield observed, as calculated by the yield formula. Additionally, the extraction conditions can significantly impact gelatin yield (*Alfaro et al., 2015*).

The concentration of NaOH and acid are critical factors influencing the efficiency of gelatin extraction from the body wall of *S. horrens*. Excessive concentrations of acid or ammonia can lead to collagen loss. Using acid and NaOH at optimal concentrations minimizes collagen loss and reduces the need for extensive washing to neutralize pH, thereby improving extraction efficiency. In this study, the concentration of NaOH used was 0.2%, while citric acid and sulfuric acid were used at concentrations of 1% and 0.2%, respectively. These concentrations were found to yield relatively favorable results.

## Physicochemical properties of gelatin and benefits

The physicochemical properties of gelatin derived from collagen of *S. horrens* were compared with the GMI standards for viscosity and protein content (Table 5). The results indicate that the acidic gelatin from *S. horrens* meets the GMI standards for food and pharmaceutical applications, as its viscosity falls within the acceptable range for tablets and food-grade gelatin. The protein content of the *S. horrens* gelatin is also comparable to the standard, suggesting it could be used in soft capsules when mixed with higher-viscosity gelatin. Additionally, this gelatin is suitable for food products that require refrigeration, as it begins to gel at low temperatures (5 °C). The acid extraction method employed is

commonly used to enhance collagen absorption. *S. horrens* has a high collagen content, approximately 70% of which is insoluble and can be hydrolyzed to valuable gelatin (*Siahaan et al., 2017*; *Bandaranayake & Rocher, 1999*; *Kim & Pangestuti, 2011*). This low lipid and high protein content make sea cucumber a promising source of bioactive peptides. The denaturation of collagen by heating breaks intermolecular bonds, converting it into gelatin (*Zhang et al., 2023*). Acid extraction further strengthens the gel, with gel strength and melting point being key physical properties. The gel strength and melting point of the gel are one of the most important physical features of gelatin. The gelling point of *S. horrens* gelatin falls within the range reported for cold water fish gelatins (4–12 °C) (*Draget, Philips & Williams, 2009*; *Gómez-Guillén et al., 2011*; *Karim & Bhat, 2008*), but is lower than that of warm water fish gelatins (15–22 °C) and mammalian gelatins (20–27 °C) (*Derkach et al., 2020*). Gel formation in gelatin is influenced by acid and alkaline hydrolysis. Gelatin, when dissolved in water, can form a thermally reversible gel. The gelling point of aquatic gelatins, which is lower than that of mammalian gelatins, is affected by factors such as amino acid composition, raw material nature, and molecular weight of gelatin peptides (*Rosli, Ahmed & Sarbon, 2023*). Extraction conditions also play a crucial role in determining the gel formation point and gel strength. For instance, using high concentrations of sodium hydroxide, sulfuric acid, and citric acid can produce gelatin with the lower gel strength, as indicated by the bloom value. The stability of the gelatin's helical structure, essential for gel formation, relies on the content of imino acids, especially proline and hydroxyproline.

Gelatin's gel exhibits thermal reversibility, beginning to melt at a critical temperature, known as the melting point. The melting point of the *S. horrens* gelatin is higher than that of cold water fish gelatins (11–21 °C) and is close to the range for warm water fish gelatins (22–29 °C) (*Derkach et al., 2020*) and mammalian gelatins (28–34 °C) (*Draget, Philips & Williams, 2009*; *Karim & Bhat, 2009*). Similar findings were reported by *Pradarameswari et al. (2018)*, who noted a melting point of 28–30 °C for Pangas catfish (*Pangasius pangasius*) skin gelatin (*Pradarameswari et al., 2018*). The gelling and melting points of gelatin are influenced by pH, ionic strength, and imino acid content (*Ranasinghe et al., 2022*), However, the reasons for the lower gel point and higher melting point of *S. horrens* gelatin remain unclear, as there is limited research on sea cucumber gelatin for comparison.

The melting point of *S. horrens* gelatin, which is below human body temperature, allows it to melt in the mouth. This property is particularly valuable for pharmaceutical and food products, such as pastilles. The melting point is a significant physical feature of gelatin, affecting its rheological properties and suitability for various temperature conditions. Sea cucumber gelatin holds diverse potential functional applications. *Zhao et al. (2007)* extracted gelatin from *Acaudina molpadioidea* and reported high Angiotensin-I-converting enzyme (ACE) inhibitory activity in the GH-III fraction (<1 kDa molecular weight, $IC_{50}$ of 0.35 mg/ml) (*Zhao et al., 2007*).

*Wang et al. (2010)* found that gelatin hydrolysate from *S. japonicus* with a molecular weight of 700–1,700 Da exhibited strong free radical scavenging properties and could inhibit melanin synthesis and tyrosinase activity in B16 cells. Additionally, sea

**Table 5 Comparison of viscosity and protein amount of *S. horrens*'s body wall gelatin with GMIA standards (*Gelatin Manufacturers Institute of America, 2019*).**

| Gelatin | Protein (%) | Viscosity (cp) |
|---|---|---|
| Acidic gelatin from *S. Horrens* | 87.93 | 2.065 |
| Food grade acidic gelatin standard | 84–90 | 1.5–7.5 |
| Soft capsules acidic gelatin standard | 84–90 | 2.5–3.5 |
| Tablet acidic gelatin standard | 84–90 | 1.7–3.5 |

cucumber-derived proteins like gelatin and collagen have significant potential as ingredients in cosmetics, such as anti-aging creams, UV protection lotions, lipsticks, and whitening creams (*Siahaan et al., 2017*). Thus, *S. horrens* gelatin could be a valuable source of bioactive compounds for the cosmetic industry, though further testing is needed.

## CONCLUSIONS

In summary, collagen extracted from the body walls of *Stichopus horrens* and *Holothuria arenicola* was identified as type I collagen. consisting of $\alpha_1$ and $\beta$ chains with molecular weights of 125 and 250 kDa, respectively. This characterization is supported by amino acid composition and SDS-PAGE analysis. Gelatin derived from *S. horrens* collagen exhibited distinct properties compared to mammalian gelatin. These findings suggest that collagen and gelatin from these sea cucumber species could serve as viable alternatives for human consumption, either independently or in combination with supplements to enhance their health benefits. Improving the extraction process could further refine these properties. Due to their low risk of disease transmission and cost-effectiveness, marine collagen and gelatin from sea cucumbers offer promising applications in pharmaceuticals, food supplements, nutraceuticals, and cosmetics. Based on this study, *S. horrens* and *H. arenicola* are recommended as safe and effective sources for producing collagen and gelatin, with potential for cultivation in the Oman Sea to support this industry.

### Funding
This work was supported by the Chabahar Maritime University. The funders had no role in study design, data collection and analysis, decision to publish, or preparation of the manuscript.

### Grant Disclosures
The following grant information was disclosed by the authors:
Chabahar Maritime University.

### Competing Interests
Balu Alagar Venmathi Maran is an Academic Editor for PeerJ.

## Author Contributions

- Noora Barzkar conceived and designed the experiments, performed the experiments, analyzed the data, prepared figures and/or tables, authored or reviewed drafts of the article, and approved the final draft.
- Gilan Attaran-Fariman conceived and designed the experiments, analyzed the data, authored or reviewed drafts of the article, supervision, and approved the final draft.
- Ali Taheri conceived and designed the experiments, analyzed the data, authored or reviewed drafts of the article, co-Supervision, and approved the final draft.
- Balu Alagar Venmathi Maran analyzed the data, authored or reviewed drafts of the article, and approved the final draft.

## Data Availability

The raw data are available in the Supplemental Files.

## Supplemental Information

Supplemental information for this article can be found online at http://dx.doi.org/10.7717/peerj.18149#supplemental-information.

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
