# Peer review of "Extraction and characterization of collagen and gelatin from body wall of sea cucumbers Stichopus horrens and Holothuria arenicola"

_PeerJ, doi:10.7717/peerj.18149_

## Round 0.1 · original submission · Major Revisions

Two expert reviewers have evaluated your manuscript and their comments can be seen below and in an attached PDF. Please ensure that you attend to all of the comments made by both reviewers. Pay particular attention to the comments of reviewer 2 on experimental design, sample size, and statistical analysis all of which must be clearly attended to in a revised version of hte manuscript. Also make sure that the revised version uses correct English grammar and spelling throughout.

Reviewer 1 ·

Basic reporting

The manuscript, "Extraction and Characterization of Collagen and Gelatin from Body Wall of Sea Cucumber Stichopus horrens and Holothuria arenicola and its Chemicophysical Properties," investigates the potential of collagen and gelatin extracted from two sea cucumber species as alternatives to mammalian sources.

Strengths:
Clear research question and methodology.
Detailed description of extraction and characterization techniques.
Identification of extracted collagen as type I.
Interesting properties of S. horrens gelatin (high melting point, low gel formation temperature).
Comparison of results with other marine and mammalian sources.
Potential applications of sea cucumber collagen and gelatin are mentioned.
Weaknesses:
Novelty of using these specific sea cucumber species needs to be emphasized.
Limited data on H. arenicola due to unavailability.
Discussion on applications could be further elaborated upon.
Recommendations:
Highlight the novelty of using S. horrens and H. arenicola for collagen and gelatin extraction.
If possible, include any available data on H. arenicola collagen.
Expand on the potential applications of the extracted collagen and gelatin, considering their unique properties.

Line 79: Consider mentioning the environmental and ethical benefits of using sea cucumber collagen as an alternative to mammalian sources.
Line 380: Consider replacing "bloom" with a brief explanation (e.g., gel strength measurement).
Redundancy: Lines 382-384 and 395-397 reiterate similar points about gel formation and melting points. Consider streamlining this section.
Discussion on Applications: While the text mentions potential applications, elaborate on how the unique properties of S. horrens gelatin (high melting point, low gel formation temperature) might benefit specific applications (e.g., high melting point for food products requiring stability at higher temperatures).
Overall, this section complements the analysis presented earlier. However, some minor revisions can further strengthen the manuscript.

Experimental design

The manuscript clearly outlines the purpose of the study: to investigate the potential of collagen and gelatin extracted from Stichopus horrens and Holothuria arenicola as alternatives to mammalian sources.
A detailed description of the methodology for collagen and gelatin extraction, amino acid composition analysis, and rheological property determination is provided.

Validity of the findings

The results indicate that the collagen extracted from both sea cucumber species is likely type I, based on SDS-PAGE analysis and amino acid composition.
The gelatin extracted from S. horrens exhibits interesting properties, including a high melting point and low gelling point. The manuscript focuses primarily on S. horrens because H. arenicola gelatin could not be studied due to lack of access to the species. This limits the generalizability of the findings to both species.

Additional comments

Briefly summarize the importance of gel strength and melting point for gelatin applications.
Combine information about gel formation and melting points of S. horrens gelatin compared to other sources. Highlight the unique aspects of S. horrens gelatin.
Discuss how the amino acid composition might contribute to the observed gel properties.
Expand on the potential applications of S. horrens collagen and gelatin, particularly how their unique properties might be advantageous.
By incorporating these suggestions, you can create a more comprehensive and impactful conclusion section.

Reviewer 2 ·

Basic reporting

- English proofreading is necessary. I found several problems with grammar and misspellings in the text.
- Some parts of the manuscript do not follow the journal guidelines. The structure is not neat
-

Experimental design

- I could not find the experimental design in the manuscript.

Validity of the findings

- The sample size is insufficient to represent the population. Only two individuals were used during the experiment
- There was no repetition in the study
- There were no statistical analyses

Annotated reviews are not available for download in order to protect the identity of reviewers who chose to remain anonymous.

---

## Round 0.2 · Minor Revisions

Thank you for submitting your manuscript. Although you have submitted a track changes version, it seems that you have simply replaced the whole manuscript as track changes. This does not allow me or reviewers to easily see where changes have been made and to see exactly what you have modified. This is the whole point of submitting a track changes version. For this reason I am asking you to modify the previous version of the manuscript to only show the modifications that were make to the text and resubmit your files.

Staff Note: When you resubmit, please be sure to also *re-upload* your rebuttal letter

---

## Round 0.3 · Minor Revisions

I have evaluated your revised submission. I am satisfied with the modifications that you have made to the manuscript but I do think that it is rather repetitive in places and could be tightened.

In addition, there are many errors with respect to the English language that need to be corrected before final approval. I am providing a few examples below but there are many more especially with missing verbs. Please ensure that the text is thoroughly reviewed before submission.

Line 2: from Body Wall should read from the Body Wall
Line 26 mollusca should read molluscs
Line 137-138: has a more thermostable superiority and a more tightly collagen structure sounds odd. I suggest … is more thermostable and has a tighter collagen structure
Line 147 abundantly should read abundant

---

## Round 0.4 · accepted · Accept

Thank you for addressing the issues that were raised. I have read over your manuscript and am pleased to recommend it for acceptance by PeerJ.